# Evaporating Metacognitive Talk: School Inclusion, Power, and the Interplay of Structure and Agency

**Ezra Temko** 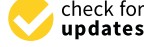

Department of Sociology, Southern Illinois University Edwardsville, Edwardsville, IL 62026, USA; etemko@siue.edu

**Abstract:** This paper addresses Lukes' and Hayward's arguments that power should be conceived as agential versus structural. My fieldwork at Mitchell Primary School demonstrated that educators and students at Mitchell were structurally constrained and enabled but also exercised agency in navigating these institutional boundaries. Not only are both structural and agential conceptions of power valid, considering their interplay moves social analyses forward—at Mitchell, teachers' otherwise-frequent metacognitive talk evaporated when their inclusion-oriented practices were more distant from institutional norms. Understanding power requires including its sources (from the individual actor to social structure) as one key dimension. Using this understanding could help educators more intentionally make conscious choices about their inclusion practices as they navigate their school environment.

**Keywords:** power; education; inclusion; structure; agency; teaching; theory



## 1. Introduction

The structure–agency debate surrounding whether to conceive of social phenomena primarily in terms of structure (with a pure type [1] of structural determinism) or agents (with a pure type of individuals having no structural constraint regarding free exercise of power) has been long-lasting, in the 1960s manifesting between Miliband and Poulantzas [2,3]. This debate resurfaced with Clarissa Hayward's 2000 [4] critique of Steven Lukes' 1974 [5] theory of radical three-dimensional power, and by extension, her critique of the other theories of agential power to which Lukes had critiqued and added a dimension. Lukes [5] and Hayward [4] both investigate forms of power that shape consciousness, ideology, and interests. Lukes and Hayward both agree that cognitive influences distort perceived preferences and societal norms and practices, and both investigate forms of power that shape consciousness, ideology, and interests. However, Lukes attributes this power to agents exercising power (who can thus be held responsible), whereas Hayward attributes this power to social boundaries that shape individuals' (including powerful actors') freedom and possibilities [6].

There is a robust body of literature exploring agency, structure, and their relationship, and scholars have explored this domain within the field of power theory; however, explicit and direct engagement of both Lukes' and Hayward's theorizing in concert has occurred but been limited [7–11]. Furthermore, while Lukes' theory has empirical works that test its validity [12,13] and Hayward [4] conducted an ethnography as part of her theorizing, both theories lack a robust body of applied empirical research [11,14], and there have not been any empirical studies that simultaneously apply and test the validity of both Lukes' and Hayward's theorizing. Exploring whether Lukes' and/or Hayward's respective claims make sense of social reality through an empirical study can help lend credibility to their theorizing and/or suggest areas that need retooling [2,15]. This paper thus aims to empirically apply and thus test the ideas contained within their dialogue by evaluating their utility for social understanding and analysis [16].

## 2. Structure, Agency, and Power

### 2.1. Lukes' 3D Power and Hayward's De-Faced Power

Power has been conceived as embedded within actors since its popular conceptualization by sociologist Weber in 1922 [17]. Theoretical debates around power evolved within this framework [18], while expanding from a focus on overt and observable decision-making [19] to include 'nondecision-making' and institutional bias [20,21], and in Lukes' 1974 book *Power: A Radical View* [5], cognitive manipulation—how powerful agents shape meaning, perceptions, and interests.

Lukes [5] did not reject more traditional definitions of power. His theory was additive—that while traditional conceptions focused on how the powerful impact interests those lacking power have expressed (manifest interests), it failed to additionally account for how the powerful impact those without power's perceived interests, including in ways that run counter to their real (but latent) interests. Lukes labeled this theory three-dimensional (3D) power—the powerful may exercise power through overt observable actions that get those without power to do something they otherwise would not do (1D), the powerful may limit the scope of the political process to issues which do not radically challenge their own interests (2D), and the powerful may exercise power through shaping meaning, perceptions, and interests such that those without power seemingly consent to actions that are actually against their real interests (3D). While Lukes' focus is on the powerful manufacturing the consent of the powerless to their own subjugation, his 2005 book revision notes that power can also be a positive force. Thus, shaping meaning, perceptions, and interests can also be a powerful tool for bringing about significant effects that may support the real interests of those subject to such power.

Lukes [5] also notes in his book revision that power is a broader concept than his 1974 book's focus on coercive power. Power's *form* (visible (1D), hidden (2D), and invisible (3D)) can be considered one dimension of power [22]. Other dimensions of power include *space* (closed, invited, and claimed) and *level* (global, national, local, (and sometimes household [23])), as integrated together by Gaventa [22], expanding Lukes' conception of power into a "power cube" that differentiates power by form, space, and level.

While 3D power is often referred to as 'invisible power', I use 'cultural power' to be more specific and avoid reproducing ability-based stigmatization [24]. I use the term 'cultural' rather than terms more synonymous with the concept of invisibility not linked to sight such as 'concealed' or 'imperceptible' in reference to what primarily characterizes 3D power—the "politics of signification" [25] (p. 64)—power shaping ideas [26].

Lukes' theory conceives of power as agent centered [27,28]. However, structures [29] also have a history of being described as having power, such as Durkheim's 1895 [30] (p. 3) analysis of "social facts" as "external to the individual, and endowed with a power of coercion". In 1975, Foucault challenged the dominant agential conception of power, explicitly describing power in structural terms, as circulating and producing subjects [17,31]. Lukes' theorizing bypasses how structures produce action [11] and does not "adequately resolve. . . the tension between conceptions of agency power and structural power" [32] (p. 128), which may be why, despite Lukes' insistence on attributing cultural power to agents, some theorists nevertheless interpret Lukes' 3D power as primarily a "structural interpretation of power" [10,33] (p. 735) and [34] or modify it to include structural relations of power [35]. This also reflects the ambiguous boundary between actors/agents and structures, e.g., while an individual human being is understood to be an actor, often so are social roles and collectives (e.g., a president, voters, the state, a particular family, a particular band, the U.S. Congress) [11].

In 2000, building off Foucault's theorizing, Hayward published *De-Facing Power* [4], arguing against Lukes' 3D power theory and more broadly against agent-oriented theories of power. Hayward instead argued that power should be "de-faced", i.e., conceived as structural. Hayward [4] (p. 12) argued that the right way to conceptualize power is in terms of how "social boundaries (such as laws, rules, norms, institutional arrangements, and social identities and exclusions)" shape all actors' agency, rather than in terms of how

*more powerful* actors shape *less powerful* actors. Hayward's work [4] is both theoretical and empirical. Hayward's book is based on an ethnography she conducted in two schools in different communities. Rather than focusing on how teachers enacted 3D power over students, Hayward focused on how teachers and students' freedoms, possibilities, and actions were shaped by power's mechanisms, i.e., by social boundaries. Hayward's theory conceives of power as systemic and structural [27,28]. While Hayward did not use the term "structural power" in this work, she now uses the label [4,36], and her work has been described as including "social structures and events" [28]. Hayward shares Foucault's [37] (p. 297) interest in interrogating the "set of rules by which truth is produced". However, Hayward's [4] theorizing markedly departs from Foucault as she (like Lukes [6]) is interested in liberatory power analysis. Hayward [4] asks us to consider how norms and institutional arrangements shape actors with practical implications for shaping a just society, whereas Foucault [38] is more interested in the genealogy of these norms and institutional arrangements. Hayward's theorizing has been critiqued for bypassing consideration of how, even with structural constraints, actors now ascribed no power still navigate those structures and take various actions that go beyond being structural dupes [7,11].

*2.2. Compatibility of Agential and Structural Conceptions of Power*

Hayward's focus on "systemic properties of power" and Lukes' focus on "agencies of power" are set up as incompatible [6,27] (p. 343). They present their analytic constructs as oppositionary, engaging in boundary work that labels the other's theorizing as not being about power, embracing the traditional paradigm that structural and agential conceptions of power are "in irreconcilable contradiction", with structural power being a "network of relations" that produce subjects (A has power over B, where A is structure and B are agents) and agential power being a resource [6,17] (p. 245) and [39]. Lukes also makes a normative argument for conceiving of power as agential, because one can hold actors responsible, whilst there are no direct means to hold structures accountable. Hayward argues back that this limits responsibility inaccurately, for example, focusing on actors' intentions rather than how they may nevertheless uphold oppressive structures [6].

In contrast, some theorists argue that structural and agential conceptions of power are compatible [7,8,10,17,28,31]. Haugaard [8] (p. 420) argues that "Power consists of a cluster of concepts, each of which qualifies as 'power'". Drawing from Wittgenstein's "family resemblance" idea, multiple conceptions of power form related but different ideas that still fit together [8,28]. Heiskala [17] (pp. 250,259) argues that instead of seeing these conceptions as "hostile alternatives" or as "one of exclusion and contradiction", they can be understood as part of an increasingly complex (nuanced) model "with complementary parts of a synthetic conception.... It is possible to approach power while working at different levels of the scale of power conceptions". Structural and agential "forms of power are not mutually exclusive; they can exist all at once" [40] (p. 811). In one manifestation of this, Digeser [31] (p. 991) conceives of power expansively, labeling Foucault's structural theory of power as power's "fourth face", arguing that "Power$_4$ does not displace the other faces of power, but provides a different level of analysis".

Some theorists also argue that conceptualizing power in both of these ways is imperative and worthwhile for genuinely understanding power's mechanisms and the means to contest it [7,9]. While some forms of power may be more dominant in certain situations [28], "power must be conceived as both a truly structural and a truly agential phenomenon" [7] (p. 376). Considering both together may reveal useful data and theoretical understandings. Boonstra [28] (p. 3) argues that to understand and change social phenomena, "it is necessary to include human power as well as to understand how this power is shaped through social and ecological conditions". McGee [10] shows how accounting for both helps those seeking to create change. McGee adapts an applied power matrix that addresses invisible power (as conceived by Lukes and Gaventa [41]); by reframing invisible power as de-faced power, McGee [10] is able to add ways actors can resist invisible power and transform systems.

Some theorists also argue that it is important to consider both conceptions of power because they are "interrelated" [40] (p. 811). Hays [42] (p. 59) argues that purely agential or structural models are intrinsically limited in their validity because of the relationship between the two, writing that "When social theorists use structure and agency as contrast terms (agency is what structure is not, and vice versa), they neglect the interconnected nature of the two". Boonstra [28] (p. 4) agrees, noting that "social scientists frequently fail to indicate how dimensions or sources of power relate to one another". Similarly, Digeser [31] critiques solely structural models as ignoring actor agency and therefore not allowing exploration of structure–agency interplay. Exploring and understanding this interplay can be useful for more accurately understanding the social world. As Bates [7] (pp. 356–357) argues, "An empirical instance should not be viewed as 'one thing or the other' in terms of either power/agency or structure, but rather in terms of (the power(s) of) both structure and agency and the contingent interaction between them". However, as noted, this is not the dominant approach in social science studies.

### 2.3. Theoretical Research Question

Hayward and Lukes [6] engage in boundary work [39] that labels the other's theory as not being about power. For Lukes, power is limited to actors; for Hayward, structures. This paper contributes to existing literature by providing one empirical test of their claim. Other theorists (e.g., [7,9]) consider structural and agential theories of power as compatible. In this case study, which/whose conception(s) of power effectively enable(s) our understanding of the empirical world? Do any deserve primacy as a tool for social analysis? Are they both able to be applied to this case study, and if so, does this enrich our understanding of social inclusion at Mitchell Primary School? Are these conceptions of power in conversation with one another? Do they relate to one another?

### 2.4. Power, Inclusion, and Public K-12 Education

This paper explores how best to conceive of power through a qualitative field study in an elementary school. I join Hayward and Lukes [6] in being concerned with social theories of power having relevance for the moral pragmatics of our society. As such, my area of focus is school inclusion and equity. Inclusion is "the process of improving the ability, opportunity and dignity of people, disadvantaged on the basis of their identity, to take part in society" [43] (p. 4). It is "engagement within a community where the equal worth and inherent dignity of each person is honored" [44] (p. 211). Equity reflects "the pursuit of equal opportunities and the avoidance of severe deprivation" [45] (p. 74). It has to do with fairness, justice, and eliminating systemic barriers to human capabilities [45]. Equity is "the quality of being fair; the creation of policies and enactment of practices that ensure outcomes are not predictive of identity or demography" [44] (p. 211). My focus on inclusion and equity in schools is holistic and not focused on one particular social identity.

Hayward's book was also set in the school environment, in which she explained how broader social context delimited educators' choices of their behaviors in two schools situated in markedly different socioeconomic contexts, with a focus on one particular classroom in each school. Hayward pushed back against the conventional wisdom of educators in the classroom holding the power in said setting; however, by only analyzing this one aspect of how power operates, Hayward misses the full complexity of power operating in school environments. This is valid for the purposes of outlining and detailing her contribution of de-facing power but leaves open the question of whether 3D agential power is also operating in the school environment.

School environments can empower students as well as reproduce and exacerbate inequalities [46]. Power is frequently included as part of school bullying [47] and as part of stigmatization studies [48], though often these studies underconceptualize power. Much of primary and secondary (K-12) education scholarship engaging meaningfully with power uses the "hidden curriculum" concept [49]. The hidden curriculum has similarities to Lukes' 3D power and also overlaps with Freire's popular education theories of conscientization

and critical empowerment [12], especially through their shared basis of a focus on cultural hegemony, another concept applied to educational studies [50–52]. Scholars have applied Lukes' theorizing [50,51,53] to education studies. Scholars have also applied Foucault's disciplinary power theorizing [54–57] to education, which Hayward [4] draws on for de-facing power. In addition to Hayward's [4] ethnography in the school environment, Cha [54] also applies a structural understanding of power to their school study, also drawing from Lukes, though focusing on Lukes' contribution of considering the power of ideological and cognitive forces rather than taking an agential perspective. While there are theoretical works that suggest bridging Lukes' and Hayward's theorizing, there is a lack of empirical educational scholarship with a focused attempt at applying both Lukes' and Hayward's theorizing.

### 3. Materials and Methods

I conducted ethnographic research [58–60] at Horace Mitchell Primary School in Kittery, Maine. I had no pre-existing relationship to Mitchell but had read about how, in the spring of 2015, a lesson aimed at promoting tolerance and acceptance of transgender students garnered national attention and controversy. This school thus seemed ripe for exploring how issues of inclusion and school climate and culture can be and are navigated, to varying degrees of success, in contemporary U.S. society. At the time of my fieldwork, Mitchell was an elementary school with 35 full-time staff members and about 400 students, located in the Town of Kittery, which had just under 10,000 residents. About one-third of students were economically disadvantaged and just over four-fifths were non-Hispanic white-only [61].

Broadly, the empirical research question for my field site research was: what is the culture of Mitchell School? More specifically: what is the school climate of Mitchell, and how does Mitchell foster inclusion and reproduce inequalities? What social boundaries are present that impact inclusion at Mitchell, and in what ways do agents draw upon and navigate these social boundaries in their climate work? These broad questions of culture and process were useful starting points for a grounded theory approach to analysis that begins by observing what emerges from observations and data collected detailing those observations [62]. Exploring school inclusion at Mitchell served as a case study for analyzing the broader theoretical questions regarding which theories of power manifest and elucidate school inclusion dynamics at Mitchell. Using the data I collected focused on this empirical research question, I investigated my theoretical research question, focused on investigating whether the constructs of both 3D and de-faced power manifest at Mitchell, and if so, whether considering them in tandem and relation matters.

The Kittery School District's policy on visitors encourages citizens to be involved and interested in public schools and authorizes building principals to establish administrative procedures for visits. Michell School's 2015–2016 Handbook stated that "Everyone is always welcome at the Mitchell School. All visitors are required to report to the school office when they enter the building. We do require ALL visitors to sign in and wear a visitor's name tag". I met with the school's principal, who photocopied my driver's license and approved, in writing, for me to come observe in the school to conduct my research.

After gaining permission for my fieldwork from the school principal and my Institutional Review Board for the Protection of Human Subjects in Research, I conducted 13 site visits at the school between February and June 2016, engaging in participant observation and unstructured interviews (informal conversations). I visited a variety of classrooms and grades, but I also visited a few classrooms more than once, as well as the same class while they were in different spaces (e.g., at a special like Art). I took jottings in a notebook and on my laptop while at the school (e.g., notes on what I observed, from dialogue to activities to interactions to how students seating corresponded with their gender presentation), then typed up more extensive notes from those jottings afterwards. The principal assigned me to spaces to visit, sometimes based on my requests for what I wanted to observe, and communicated about my visit to the educator responsible in that space. When I arrived at

the school, I would sign in and secure a visitor pass, a sticker I wore while at the site. While I was forthright about the purpose of my visits, educators regularly assumed I was a student teacher or training to be a teacher, and students often assumed I was a parent, as these are more common reasons for an outsider to be in the school. I corrected misperceptions, but they nevertheless contributed to a general ease others had with my bodily presence, which was not marked as out-of-place. Pseudonyms are used throughout *Section 4. Results*.

I initially began coding for "fostering inclusion". An inclusive school environment "foster[s] a sense of belonging through a culture of dignity" [44] (p. 211). Fostering inclusion includes instances in which cultural norms or dominant conceptions related to structural inequality are challenged. However, this code had low inter-rater reliability—it was too broad, vague, and open to varied interpretations. Separately, I began noticing structural and agential forces at play in my data and began explicitly coding for instances of power manifesting, using separate codes for 1D power, 2D power, 3D power, and de-faced power. While these dimensions of power are not discrete and their constitutions rely on one another, they can be analyzed separately [63]. However, these codes were also too broad and ambiguous, leading both to too many instances that were not particular enough to be analytically interesting as well as low interrater reliability (e.g., an argument exists for almost any instance to be or not be de-faced power).

I revised my coding by considering the intersection of these two areas, coding for each form of power, but only for instances that included an explicit impact on inclusion. For example, my revised de-faced power code, 'structural boundaries that impact inclusion', attended to school structure and external social boundaries that impact fostering inclusion. Both the constraint and its impact had to be evident in my field notes, and not just theoretical. For example, I did not include instances when someone mentioned that Kittery is wealthy, or has a military presence, because the impact on inclusion was not explicitly observed and written in my notes. Similarly, an observed impact had to be specifically about inclusion. Examples of instances that fit within these and other codes can be found within *Section 4. Results*. Two additional codes that emerged from my data were unchecked noncompliance (instances in which there was an explicitly stated expectation, rule, procedure, or direction that a student or educator did not comply with and that was either tacitly allowed, ignored, or unnoticed, not followed with an observed sanction) and the military.

Finally, I created an interaction of 3D and de-faced power to analyze their intersection, coding for exceptional agential instances of fostering inclusion that involved substantive pushback against social and institutional social stratification boundaries. Educators exercising 3D power was less interesting analytically when their actions more or less matched the social boundaries one would expect them to follow. By considering specifically agential actions that involved substantive pushback against social and institutional boundaries, I was able to consider and examine both agency and structure in concert, at their intersection. Specifically, I coded for instances where norms around social stratification were challenged beyond what is culturally rewarded. These instances had to involve pushback against institutional boundaries, such as challenging exclusionary notions or negative cultural mythologies relating to stigmatization or social identities, and the pushback had to be transgressive, not simply impressive. This coding also generated a residual theme: instances of 3D power that were relatively normative practices within the everyday school context.

Through interrater discussion and evaluation, I was able to tease apart the cases in which an agent acted in a way someone would likely find surprising or striking given the context versus what was actually more commonplace or frequently occurring within a particular cultural context that was more surprising to me or someone else at an individual level. This enabled me to refine my coding scheme and achieve a high level of interrater reliability. The following examples of instances from my field notes that were not coded as exceptional instances of fostering inclusion may help explain this code. U.S. public schools tend to at least superficially support racial diversity; even in a majority white school, it is not surprising that the welcome sign for the school had kids' faces of different colors. Similarly, school programs relating to their military children's population, even if

particularly impressive, are not necessarily surprising given the context that the school is situated in a community with a naval shipyard. I did not code instances of a teacher giving a boy student a (traditionally feminine) pink pencil with hearts or of the boy boasting to another student about the gift. While this may seem like it goes against gender norms, it seems fairly institutionally accepted within many U.S. public schools that students of all genders should be able to prefer whatever color they want, and that a teacher can and should support that, especially for kids. The use of the term 'criss-cross applesauce' instead of 'Indian style' is now common throughout elementary schools and not transgressive or likely even always intentional. To fit this code, instances had to be atypical; they had to go above and beyond what one would normally expect in normative school attempts to foster inclusion. While my reference for this category included my experiences with other K-12 schools, including as a former K-12 teacher, exceptional instances were also exceptional within my observations at Kittery, meaning they were uncommon and not something I observed repeatedly across the school environment.

While the final iteration of each of my codes had specific conceptualizations, including definitions of how they were to be operationalized with contextual examples, coding with these themes is nevertheless inherently context-specific and would have marked variation throughout both the United States and the world, especially depending on cultural normative educational values and practices within various spaces. My coding is based on my perceptions and experiences having studied social stratification, gone through teacher education programs, and taught in K-12 schools in the northeast. My experiences, socialization, and positionality have influenced and focused what I noticed and classified into these various codes. My coding is also based on interrater reliability with three peers, one from southern Maine and two from different geographic areas in the northeast, one from natural resources and public policy and two from sociology. I would give a peer a de-identified section of my field notes along with an operationalized code (e.g., title, definition, example), and we would both code the same section of field notes for the code. We would then identify which instances we coded in common and which instances we coded differently. This especially helped clarify which instances were exceptional and departed from conventional and common practices of fostering inclusion by ensuring I was not identifying these based on my own biases and assumptions. It helped that one of the raters had an insider perspective of community norms of the school's geographic area, balanced and complemented by outsiders to the area. Furthermore, having raters with backgrounds in education, sociology, and outside these disciplines aided in what was essentially a triangulation process that improved the validity and reliability of my findings. Besides enriching my analysis, this also led to further refining and clarifying of coding schemes, which, when improved, meant that the next time we each coded a section, there was more agreement on which instances we did and did not highlight as matching the code. While the specific empirical validity of my findings in terms of classifying instances may be questioned due to this subjectivity, this limitation primarily suggests that the findings apply with some caveats to this particular case study within this particular context and are not necessarily generalizable. Furthermore, my coding was grounded in my actual observations. For example, transgressive inclusionary practices only included practices that were uncommon. If an instance re-occurred regularly or fit into a type of instance that I commonly observed, it was not coded as transgressive for that school environment. Additionally, the theoretical findings and implications regarding whether and in what relations fostering inclusion manifests in terms of various forms of power translates and extends beyond my positionality and potential disagreements over contextual coding classifications, lending validity to my broader findings and their implications [16,64].

Instances of transgressive inclusionary practices shared some common characteristics; however, based on my data, I could only speculate as to why they occurred or what led educators to act in particular ways in these instances. To learn more about what was going on with instances of transgressive inclusionary practices, in the fall of 2022, with approval from my Institutional Review Board, I followed up with five educators who, via written

communication and/or interview, provided additional context and perspectives solely focused on my observations within this theme [65,66]. These interview participants first received a research participant notification form and consented to participate in the study.

## 4. Results

In this section, I summarize my findings by presenting observational data of power operating at Mitchell School, particularly as it relates to school inclusion. I first explore agential power: direct observable power, nondecision-making power, and cultural power, all of which manifested at Mitchell. I then present my findings related to structural power, which was also present, and finally related to the intersection of cultural and structural power, which revealed empirical insights that would have been absent without consideration of both forms of power.

### 4.1. Agential Power

The examples included here of 1D, 2D, and 3D power generally focus on educators moderating inclusion through exercising power over students; however, I also observed these forms of power among different statuses.

### 4.1.1. Direct Observable Power

As to be expected in a school environment, direct, overt displays of (1D) power were frequent and easily observable at Mitchell. Teachers regularly directed students to take or not take particular actions and redirected students or otherwise gave them consequences (related to matters of inclusion). Educators exert overt power over students, getting students to take actions that students would not otherwise take. For example, my very first visit was to Mrs. K's Kindergarten classroom. Jake was the class star and therefore got to choose the class's *Go-Noodle* game for the class to get out their wiggles. He wanted the Christmas one, but Mrs. K, said, "No, we're not going to do the Christmas one". Jake therefore chose a different one. During the game, another student, Ryan, had his head down on his table. Mrs. K told Ryan he was part of the class and had to participate, so he did. U.S. public school institutionally sanctioned celebrations focused on Christian holidays rather than equal weight being given to a multiplicity of traditions renders Christmas as normal, universal, and unremarkable, promoting one dominant cultural and religious group's beliefs and practices and othering non-Christians' identities, experiences, and cultures. As a consequence of Mrs. K's exercise of 1D power, the class participated in a game that included all students across varying potential religious and non-religious identities.

### 4.1.2. Nondecision-Making Power

In terms of 2D (nondecision-making) power, educators narrowed students' scope of choices by presenting them with particular limited choices. For example, students engaging in reading select books from those available to them. In Miss Vercelli's second grade classroom, there was an elevated reading area with books—three books appeared to be explicitly about culture or identity, two *Little Christmas Elf* books and one *Hiawatha* (a precolonial indigenous leader) book. Another bookcase on the main floor of the classroom was filled with children's books about teeth (loose teeth, losing teeth, the dentist, etc.), among them the award-winning fictional picture book *Doctor De Soto* about a mouse dentist who treats a (mouse-eating) fox who has a toothache. This pre-selection bias was also evident in educators embedding their directions and questions with assumptions of particular ideas that were presented as given, mobilizing challenges to it off the table. For example, when first grade teacher Mrs. Jones spoke with students, she started from the position that "we have to show respect". When guidance counselor Ms. G started a lesson in a third-grade classroom by holding up the book *Trouble Talk* by Trudy Ludwig by asking the class "How can we stop trouble talk?", Ms. G was delimiting the conversation to being about techniques to stop trouble talk, with the implicit assumption that stopping trouble

talk is a worthwhile goal. There were numerous other examples of actors presenting a limited scope of choice to other actors.

### 4.1.3. Cultural Power

In terms of 3D (cultural) power, Mitchell School had both an open and hidden curriculum [49] aimed at fostering norms of inclusion through the exercise of cultural power. Educators at Mitchell regularly worked to shape students' ideas about what is right and wrong and how they should interact with one another and the broader world, e.g., through signage, rules, actions, and statements.

Like many schools, Mitchell is a highly structured environment, filled with signs, symbols, rituals, routinized procedures, and directives. Many signs explicitly state values the school was trying to impart. In the art classroom, the words BE KIND were written big with one letter on each cabinet drawer. Class rules were posted such as "Treat others with kindness" and "I will share. Sharing is caring". There were hallway bulletin boards on "Kindness and Respect" and printed posters on "Fair Ways to Play". The school welcome sign featured four racially diverse children. These signs represent agential power because actors posted and created them, causing them to be there.

Part of Mitchell educators' work is shaping students' dispositions and the broader classroom climate. Educators took actions and made statements to shape students' cognition related to inclusion and who they should be and how they should act as members of the school and broader world community. Some statements mirrored signage, such as when Miss Vercelli redirected a student by saying, "Sharing is caring; we've talked about it". Often educators both named a norm the student should practice and explained why it was important. For example, in Mrs. K's class, students were sharing pictures they brought in from home. Jake (the star) went first, and he called on Sam and Mike for questions. Mrs. K commented to the class that next time, sharers need to ask others, not Sam and Mike, so that everyone gets a turn. Ten more students shared until it was time for recess, during which Mrs. K followed up on that comment in her actions, establishing taking turns and sharing space for contributing as normative practice. Mrs. K worked towards shaping students' ideologies around social interactions throughout the day. Earlier in the day, when a student brought up beating or being beat in a game, Mrs. K responded that while "playing games often involve winner and losers... it's not about beating all the time"; rather, it is about playing the game itself. In another case, after explaining their assignment, Mrs. K asked students what they should do if they are having trouble. Jon responded that they can ask for help or work on a different part that they can do. Mrs. K paraphrased this back to the class and said that if they are having trouble, "That's okay. We help each other".

The work environment at Mitchell, in relation to equity, was also in part an outcome of 3D power. For example, one teacher expressed that Mitchell's principal had communicated to them about the importance of their family in a way that made them feel Mitchell was unique and special in terms of honoring and valuing employees for taking care of their families, and even expected them to prioritize their family needs over their work. The teacher expressed that they felt empowered by Mitchell's principal to take sick days when their own children were sick, and that his reassurances about this lowered her anxiety about and helped her feel ready to return to teaching after having children. The principal's actions created an environment where educators felt that not just their work product, but their own lives and families were valued.

### 4.2. Structural Power

I also observed de-faced (structural) power, with over 100 coded instances in my fieldnotes of structural boundaries that impact inclusion. The community and school context—Mitchell School is located in Kittery, a relatively rural community with a military (naval shipyard) and tourism (coastal town) presence—had particular impacts on inclusion. External social contexts also mattered, from federal and state laws to external norms and microaggressive practices students and staff brought into the school setting to resources

that exist within broader society (e.g., materials, curriculums, websites, reproducibles, books, etc.) that educators draw from, including many designed for them, which come with their own sets of hidden curriculums and biases.

The size of the school, the community it is located in, and its physical environment all impact inclusion. Located in a rural New England town, Mitchell is a small school. An educator shared that this makes it "really hard to be invisible here", and, thus, students get enveloped in a "culture of nurture and caring". However, an educator also shared that the lack of racial diversity in the community can sometimes lead to unintentionally ignorant racist comments or actions. An educator also noted she cannot teach kids not to touch guns because about half the community likely have guns in their home and some third graders are learning to shoot or hunt. She therefore tailored her teaching to be inclusive of community practices. As an example of the school's physical environment impacting inclusion, classrooms at Mitchell with single bathrooms within the classroom shape those bathrooms being all-gender.

The military produces institutional arrangements, norms, and identities that have a substantive impact on inclusion at Mitchell and on how actors navigate inclusion at Mitchell. Kittery is home to the Portsmouth Naval Shipyard, one of only four naval shipyards in the United States [67]. According to one of the guidance counselors, every class has at least three or four military kids (excluding civilian children whose parent(s) have jobs associated with the military). Over one-fifth of students at Mitchell are military connected [61]. The military came up during my observations over two dozen times, usually during informal conversation, especially when I asked people about the school. One educator, when asked about the biggest challenge the school faces, immediately responded that it was the "military population". Another educator and parent, when asked what she thought about the school, first responded with her concerns about the resource drain of military kids on the area schools. A teacher, sharing some context with me about her classroom, highlighted the transience of students moving in and out of the school and her classroom as a result of the military. More than one teacher mentioned that military kids are sometimes in three different schools in one year. According to Mitchell educators, military kids' lives "can be traumatic" and can have social challenges (e.g., a military kid spit on another student the day after their father was deployed) and academic challenges (e.g., a new transfer military student only knew one-quarter the number of sight words most of the rest of the class knew). This community and school context impacted inclusion at Mitchell and what and how actors navigated issues of inclusion.

Being on the seacoast lends itself not only to having a naval shipyard but also to the town being an active tourist destination. School approaches to diversity and discipline are influenced by the community context, in this case Kittery being simultaneously a military community and a tourist destination. About an hour drive north of Boston, Massachusetts, Kittery is literally halfway over the bridge from the popular town of Portsmouth, New Hampshire, and has its own downtown, retail outlets, and beaches and forts. In terms of student behavior and discipline, I rarely observed students engaged in continued defiance or high-level deviant behaviors characteristic of many school environments. However, despite all the structure and rules in place about what students were supposed to do, the school culture includes students regularly engaging in minor behavioral infractions without observed sanction. I observed over 50 instances of this unchecked noncompliance, which mirrors the larger cultural context of the town, a straddling of Kittery's liberal and inclusive, playful tourist face and its traditional and conforming regulated military and rural face. There was an ambiguity and situatedness to what rules were actually required and enforced in what settings. Students negotiated their adherence to disciplinary structures through sometimes complying and sometimes not complying. Mitchell was neither a school that was extremely strict, where every instance of non-compliance was met with a redirect and consequence, nor a laissez-faire environment, in which students did what they wanted without admonishment or intervention. Throughout each classroom,

even one recommended to me as more "structured", there was negotiation, straddling, and sometimes tension between these pure types.

Mitchell culture is necessarily informed by U.S. and Maine educational laws and regulations. For example, the structure of the U.S. education system is also one of "inclusion" and special education needs coming with legal obligations [68]. Reflecting this, there were many special education teachers and paraprofessionals at Mitchell who provided additional services to students, enabling them to be in the classroom, e.g., a one-on-two who worked with an autistic student and a behaviorally problematic student.

External norms from the community and broader society were brought into and enacted within the school environment. For example, when given choices, students mostly self-segregated by gender for seating (generally with girls sitting in the front and boys in the back), partnering, and calling on others. Student color choices (e.g., for clothing, for balloons during a P.E. activity in the gymnasium) were also dominated by only girls wearing or selecting 'feminine' colors. I repeatedly observed educators ask children about their mom when, per the kids' correction, the appropriate caregiver to ask about was actually the dad. The school's organizational personnel structure also reflected broader gender norms, with a man principal, all women teachers, and all men custodians. The earlier controversy over the school's gender inclusion lesson using the book *I Am Jazz* about a transgender girl initially began because some cissexist parents objected to the lesson. However, multiple educators shared that, overall, the national attention did not translate to a controversy within the school—that all the students got it, with perhaps the exception of "a few third graders who brought in other ideas from outside".

Educators and students' decisions and actions took place within and were shaped by societal, community, and school structures. Educators used many already existing resources (materials, curriculums, websites, reproducibles, books, etc.) designed for them, e.g., *Second Step*, *GoNoodle*, *MotherGoose.com*, and *Everyday Mathematics* were used at Mitchell. These resources are laden with particular values; they come with their own hidden curriculum and biases. For example, a teacher played a *YouTube* video from *Harry Kindergarten Music* called "If You're a Kid (Dance Around!)" in their classroom, which at times directs students to participate in particular activities based on being a boy or a girl. Music played in P.E. class came with particular messages, e.g., *All About That Base* has sexual overtones. The guidance counselors made use of numerous books to promote understanding of race (e.g., *Skin Again*), military children's experiences (e.g., *The Night Catch*), and social skills and perspective-taking (e.g., *Red: A Crayon's Story*; *The Skin You Live In*).

### 4.3. The Intersection of Agential Cultural Power and Structural De-Faced Power

While structure certainly played a prominent role in shaping school inclusion, it was not uniformly navigated by the school's actors. Through interrater discussion and evaluation, I was able to tease apart cases in which an agent acted in a way that was more commonplace or would likely be institutionally and culturally rewarded within the school's particular cultural context versus actions that were more surprising, striking, or transgressive within that context. Most of the instances of educators working to foster inclusion were ones that are relatively normative for U.S. schools that support multiculturalism, with corresponding practices that would be institutionally and culturally rewarded. However, there were also nine exceptional instances of fostering inclusion in my field notes—transgressive practices that departed from more common actions.

By analyzing this intersection of agential and structural power, I was also able to draw an insight that was not apparent to me prior to this specific analysis. Normative educator practices of fostering inclusion commonly involved metacognitive talk. For example, in Mrs. Pine's Kindergarten class, students were brainstorming potential activities for a special class reward they had earned. When a student proposed a "Fancy Nancy Day" party, three boys expressed disagreement with the idea, two giving it a thumbs down. Mrs. Pine then intervened, sharing with the class that "When people brainstorm, it's not okay to say someone's idea isn't good" because that can hurt their feelings. In another case, a teacher

followed up their directive of "Hands down while I'm talking" with an explanation that this was important because if student hands are up while she is talking, the students are going to be thinking about "what you want to say, not what I'm saying". Throughout the teaching I observed, it was commonplace for educators to provide such explanations. Even if explanations had occurred earlier in the school year or at times I was not observing, they were also provided and/or reinforced during my observations.

However, in the instances I observed of transgressive educator practices of fostering inclusion, the metacognitive rationalization disappeared. Multiple terms were casually used at Mitchell to refer to groups, such as "boys and girls", "everyone", and more frequently, "guys". While commonplace phrases, "boys and girls" reproduces a gender binary and "guys" uses man-centered language as universal. In contrast, one educator frequently used the term "friend" to refer to students, e.g., saying "Hello friends" during her lessons. In later conversation, I asked about this term, and she shared that she uses it both "to be gender-neutral" and as a way to discuss bullying (e.g., "how to be a good friend") without using the term "bully", which she said has a lot of baggage. In another instance, when a student asked their teacher if the class could play hangman, Miss Vercelli responded, "We're not going to play Hangman. I don't like to call it Hangman". A girl followed up, asking if they could play Flower, and the teacher responded, "We can play Flower. I like to play Flower". Hangman is a game where players try to guess a word, one letter at a time, and for each letter guessed that is not part of the word, a part of a hanged stick figure is drawn on a gallows. Players win if they guess the word before the stick figure is fully drawn.

During another observation, Mrs. K's class was making a King Cole from construction paper. This corresponded with the British nursery rhyme *Old King Cole*. Popular depictions of the nursery rhyme construct King Cole as pink skinned. Mrs. K was showing students how she made her figure and noted that for the face, "I simply used a brown paper", and then she moved on and asked the class about what kind of shape she used for another body part. The teacher showed the class how to use one of the math manipulative circles to trace and cut out the face. The teacher's King Cole, and all the students' King Coles, had a brown face. No one asked about this or made any comments about it. In another instance, Mrs. K's students were getting out their wiggles to the song, "If You're a Kid (Dance Around!)". While mostly boys did the designated boy activities (e.g., "If you're a boy, fly a spaceship") and mostly girls did the designated girl activities (e.g., "If you're a girl, jump rope!"), it was also very common for boys to join in for designated girls' activities and vice versa. When boys or girls participated at the 'wrong' times, no one (teacher or student) said anything or intervened, positively or negatively. In each case I observed educators fostering inclusion through less normative choices, they did not elaborately justify their choice or explain its meaning to students. They simply acted and moved forward.

The educator who used the term "friends", reflecting on why she did not explain why she used the term, shared she felt there was no need, noting educators do not explain why they call someone buddy or sweetie. On the rare occasion someone asked, she would respond that she felt "friend" helps everyone feel included. In both of these cases, part of the provided reason for not engaging in metacognitive talk was to normalize the act. Miss Vercelli had training in teaching metacognition and was committed to students knowing why they were doing things. While she had reason for not playing Hangman—she was uncomfortable drawing parts of persons being hanged (but comfortable drawing flowers), she reflected that she simply introduced the Flower game, without explanation that this was an alternative to Hangman or why they were playing this game instead, in part because she wanted to keep her student interactions positive and upbeat. She shared that students familiar with both games made the connection themselves. While multiple educators were not aware of the game Flower being played in the school, as of 2022, multiple teachers play it at Mitchell, and while some teachers do not provide explanation, at least one teacher had shared with students that they play Flower because the other game is hurtful.

Educators did not always have explanations for why they did or did not provide explanations. Mrs. K said that, similar to when she introduced the King Cole activity, when students do self-portraits, she gives students choices for their materials, but does not comment why. While I did not observe this during the wiggle activities while I was present, Mrs. K also shared that she had previously prefaced wiggle activities as being about getting blood flowing for learning and not about following directions, so while she had not discussed gender regarding the activities, she had explicitly told students to do whatever actions they liked for all the wiggle activities they did. This provided a challenge to the institutional boundaries the song otherwise set. Mrs. K also reflected that their classroom context was open and inclusive when it came to gender expression, something she had learned about and improved upon over her years teaching, e.g., through an experience with having a trans girl student who transitioned name and dress while in her class.

One reason educators may more consistently use metacognitive talk for more normative practices is that educators may be more hesitant to explicitly discuss matters bound up in controversial cultural politics, either because they are nervous about reactions or because they have not developed the developmentally appropriate language to share their rationales (educators may first need to be aware of and clarify their own thoughts on their practices and then translate that to how they will communicate them to children). When I observed in 2016, guidance counselors were doing push-in guidance lessons anchored by books they read to the class. As of 2022, classroom teachers are leading these. It was shared with me that, while there was some initial hesitation, teachers had support from their administration, underwent years of training focused on equity and inclusion, and after doing it, got comfortable with it. One educator shared that with that increased comfort, if I came back and observed in 2022, I would likely hear more open talk and explanations on equity topics.

As of 2022, it was shared with me that Mitchell educators still have difficulty and discomfort navigating political culture around equity topics, such as having conversations with families who are not always supportive or in some cases pull their child out of certain lessons. Mitchell educators still do equity work, but proceed with caution, and have stepped back a bit due to the political climate, including angry and upset parents, which can overwhelm educators who are also trying to manage the rest of their already full jobs, which has also included navigating the COVID-19 pandemic. When I observed at Mitchell, a *Teaching Tolerance* poster "Know the Difference" was up on the wall in the guidance room, which explains sex, gender identity, gender expression, and sexual orientation. One counselor shared that it was there primarily as a symbol of welcomeness. It was not frequently used or discussed, but besides what it stood for, it served as a resource for when relevant issues arose. The poster is no longer up, but similar materials (e.g., the gender unicorn and genderbread person) are still available to be pulled out and serve as resources for particular conversations. Mitchell educators sometimes will not explicitly bring up matters of social identity, but instead follow the lead of students. For example, if they are reading the book *Red*, which is explicitly about a red crayon who has a blue wrapper (but has been banned for its potential to teach acceptance of gender diversity [69], if a child relates to the story and brings something up around social identity, they will discuss it. Mitchell educators are nevertheless also reading books that are more explicit, such as *they*, *she*, *he easy as ABC* by Maya Christina Gonzalez and Matthew SG about pronouns. With Mitchell's Civil Rights Team (a state project aimed at reducing bias in schools [70]), educators have parent permission to talk about social identity openly, and, in turn, educators feel more comfortable doing so.

This limited exploration of why metacognitive talk evaporated in these contexts revealed that actions were sometimes meant as background resources, as symbolic displays of inclusion and to provide an open door for students to enter if there was something they wanted to talk about related to it. Sometimes hesitancy existed because of the political climate or not wanting to experience reactions from angry parents. Other times, educators wanted to normalize what they were doing and not provide additional justification or

explanation they did not feel they would provide in similar but more normative instances. Other times, they did not know why they did or did not use metacognitive talk, or they may not have had a ready understanding and language to communicate rationales in a child-centered and intentional, thoughtful way. Regardless of its determinants, during my observations, metacognitive rationalization, so prevalent at other times, was absent from all instances of transgressive educator practices of fostering inclusion.

## 5. Discussion

### 5.1. Both Agential and Structural Power Exist

While Hayward's and Lukes' theories are set up as mutually exclusive, this case study provides evidence that 1D, 2D, 3D, and de-faced power are all robustly in operation at Mitchell School. Actors such as educators exercise power and teach a hidden curriculum, while institutional norms shape their actions. Given this, the choice between Lukes' agential and Hayward's structural power concepts presents a false and unnecessary dichotomy.

The development of the second face and then the third dimension of coercive power were all additive (Lukes [5] added to Bacharach and Baratz [20,21], who were adding to Dahl [19]), but Hayward's [4] critique attempted to negate rather than add to Lukes' theory. While Hayward and Lukes [6] present convincing arguments for why it is useful to conceive of power within their respective frameworks, and certainly not all conceptions of power are compatible with one another, they each fail to offer convincing arguments as to what is lost by conceiving of power using both of their respective frameworks in concert, especially when both fit into the broader definition of power as the ability to bring about an effect [5]. Their "perceptions of power are [unnecessarily]... in mutually exclusive competition" [8] (p. 420).

"When an agent makes a difference in the social world, they have power" [63] (p. 153). At Mitchell, educators were powerful agents in the classroom. However, they were also shaped, constrained, and enabled by various institutional contexts and social boundaries. Hayward's theorizing has been critiqued for equating "structure" with "power" and thus reducing the meaning of the concept of structure [11]. In this case study, power is not synonymous with structure, nor with agent or actor, but is instead a property of actors/agents and structures. Actors such as educators and students all had power capacity and exercised (constrained) agency in their actions; structures such as community, school, and external contexts, including laws and norms, all had power capacity and had (constrained) impacts on actors. Hayward's contribution of de-faced power is useful, but conceptualizing it and operationalizing it as an additive rather than rejective understanding of power helps make sense of how power truly operates in the empirical reality of the world, where both people and systems exist, are interdependent, and "yet neither can be reduced *to* the other" [71] (p. 17).

### 5.2. Considering Both Agential and Structural Power, and Their Interplay, Matters

This case study provides evidence that considering structural power, agential power, and their particular interplay can lend unique insights to analyses of phenomena and processes. Structure and agency do not exist in isolation; they continuously produce one another [28,71]. Actors are constrained (and enabled) by structural power [4,29,30], but, likewise, structures (and thus structural power) are determined by the actors who continuously intersubjectively produce them [63,65,72]. This ongoing "structuration" [73] has implications for moving power theorizing forward [10,28,63]. At Mitchell, an interactional analysis of agential and de-faced power in relation to the social world of school inclusion revealed that, while within more conventional social boundaries educators frequently engaged in metacognitive talk, educators' metacognitive talk evaporated when their actions were less normative. In providing explanations in particular types of cases and not others, educators are (re)producing structures and institutional boundaries for their school environment, just as those boundaries are influencing their actions.

Actors navigate context, which shapes and guides possibilities and actions. They decide what to do through an interpretive process that "hinges on an interaction between

individual orientations and situational constraints" [74] (p. 203). When expectations are clear, actors typically act in predictable manners; less is known about what actors do when faced with ambiguous expectations [74]. Schools are ambiguous multi-faceted sites that are part of larger complex institutions. They have many social boundaries that exist and coerce, yet which can also be permeable and fluid. Issues of climate, culture, and norms vary across contexts, and where the line is drawn around appropriateness and convention is based on socially constructed, culturally contextually specific notions. Some conventional practices at Mitchell are transgressive at other schools and, likewise, some transgressive practices at Mitchell can be conventional elsewhere. Norms at Mitchell also change over time and in response to various stimuli; e.g., after years of professional development focused on inclusion and equity, Mitchell teachers in 2022 feel more confident and comfortable discussing these topics, and now lead the read-aloud lessons guidance counselors used to lead within classrooms. Considering individual action situated within particular structural contexts offers insights into individual action but also more broadly into understanding processes that may extend in similar ways beyond the individual case [29,64].

5.2.1. Implications for Inclusive Education

When educators acted transgressively to foster inclusion, I observed an evaporation of metacognitive talk. Educators have power in the classroom, and they exercise it. For normative instances of fostering inclusion, educators included metacognitive talk, a pedagogical practice they frequently engaged in throughout the school day. However, during my observations, when educators' practices were less common and departed from normative school practice, educators reacted to this context through engaging in their transgressive practice without giving context or explanations to students like they otherwise would.

This absence of metacognitive talk is a strategic reaction to comply with the given institutional order [4,37]. This may be intentional, subconscious, or a mix. There were no explicit rules or regulations against Mitchell educators engaging in metacognitive talk in relation to these practices. However, control can operate in coercive ways through prevailing disciplinary structures and discursive practices. Institutional relations operate through educators who embody them, reproducing and legitimating them through producing explicit knowledge claims for some practices and discursive silence for others [75]. These sets of rules, however, can be relatively plastic. If educators, rather than accepting and internalizing the rules, interrogate and analyze the rules and their relationship to these rules, they can also contest and modify them; they can actively choose how to participate in them [37]. While constrained, educators still have agency in how to navigate institutional structures, though they may be unaware of how their thoughts and actions have been shaped by dominant actors and institutions [5]. For educators to actively choose whether and how to use metacognitive talk about less normative school inclusion practices, they need to be able to be intentional and to consider and navigate matters of strategy, safety, communication, and consciousness. This requires their awareness of this process of evaporating metacognitive talk, and of its contours. Empowering educators to use metacognitive talk more often in these situations would require changing the institutional boundaries impacting them, for example, by equipping educators with increased multicultural competency, space to process, language to discuss issues of inclusion and equity, and institutional support for doing so.

More broadly, considering structural power, agential power, and their interplay invites sociological questions about educational issues that focus on this interplay and consider educators as actors within diverse structures. For example, how do educators address bullying in school settings with racial conflict? How does bias interrupter (i.e., 'upstander') intervention operate differently in supervised and unsupervised settings? What institutional contexts create environments where classroom teachers feel confident and empowered to move inclusion forward, and how can various educators help shape that environment? I did not have any intentional focus on educators' metacognitive talk when I went into my observations; this was something revealed through data analysis. Future studies that begin

with an inquiry into and attention to educator's metacognitive talk could be useful to learn more about relevant dynamics and determinants.

5.2.2. Implications for Theorizing Power

On its surface, the utility of both 3D and de-faced power indicates that conceptualizing power using both concepts would aid our understanding of the social world. One option could be to consider Hayward's theory as additive rather than negating Lukes' theorizing. Expanding the concept of power to include agential and structural understandings would be a positive forward move for operationalizing power, as it would better contribute to our understanding of and interactions with the social world. This has already been attempted by some scientists and practitioners. Preceding Hayward's work, Digeser [31] added a "fourth face" of structural power to 1D, 2D, and 3D power. In political science food literature, Baker et al. [76] (p. 740), while not engaging directly with Hayward [4], conceptualize power by replacing 1D, 2D, and 3D power with three forms of power: "instrumental power", which is actor-centered, "discursive power", which is also actor-centered but "can also manifest in more structural forms" and "structural power". In this study, I similarly attempted to use an additive model, though with explicit engagement of both Lukes (like Baker et al. [76] and Digeser [31]) and Hayward. In contrast to Gaventa's [12] study focused on applying 3D power and Hayward's [4] study focused on applying de-faced power, I used both concepts of power, and determined through my empirical observations which applied to my case study and how it did so. This enriched my analysis and understanding of the school climate and culture at Mitchell. Including both 3D and de-faced power, or otherwise conceptualizing both an actor and a structural-focused model of power, is a positive move forward that could be replicated in other studies and applied work.

While power dimensions can be analyzed independently, they co-occur simultaneously and impact one another. They do not exist in isolation [63]. While I coded my data focused on the dimension of power most evident in that instance of my data, other dimensions were nevertheless simultaneously informative and present. Forms of agential power are inextricably tied together, just like structure and agency. All interactions necessarily involve all four dimensions, and considering their interplay improves our understanding of social reality. For example, the teacher who expressed the principal's support for taking sick days when their children were sick exercised 1D power by taking a sick day when they needed it. However, they are compelled to act within a context of constrained perceptions and options as a result of 2D power, influenced, for example, by daycare and school policies about their child's ability to attend, their childcare-related options and caregiving support networks, and their child's expressions of how they feel and of their needs. Teachers at Mitchell also made their decisions, with constrained options and perceptions, within an environment where the principal had shaped cultural ideas through the exercise of 3D power, communicating authentically to teachers that "family comes first", before work, constructing an environment where teachers did not feel guilt for taking off work when their kids are sick, which influences decisions teachers then make, such as taking off work and communicating honestly about their leave. This operates within a de-faced power environment where federal and state laws, along with collective bargaining agreements, define the scope of what teachers can legally do and what principals can legally enforce, as well as within a society with individualistic childrearing practices. Laws and rules around this are, in turn, influenced by actors' perceptions, experiences, and advocacy.

Lukes [5] made an important contribution with "invisible power", engaging in a sociological tradition of investigating the production of phenomena beyond the stories presented by their taken-for-granted "facades" [77] (p. 31) and [78]. Exercises of 1D power are enabled and constrained through 2D and 3D power. Similarly, Hayward [4] made an important contribution by demonstrating how power theory and analysis is incomplete without referencing structure, engaging one step further in this same sociological tradition, moving from the institutional structures and ideologies produced by the powerful [79,80] to underlying structural determinants that influence both the powerful and less powerful.

However, just as an understanding of student hunger would be undertheorized without attention to the role of neoliberalism in producing hunger, it would not make sense to ignore how social actors frame these issues and put forward policies that impact hunger and its contexts, or to ignore actual service delivery and use of programs like free and reduced school breakfast and lunch. While power theory and analysis focusing only on agency without structure [5] is insufficient, power theory and analysis focusing only on structure without considering actors' agency [4] is similarly inadequate.

*5.3. Conceptualizing the Multiple Interactive Agents of Power: The Sources Dimension*

While a tidy improvement, simply adding de-faced power as a fourth dimension of power is conceptually clumsy. Attempts to combine or otherwise include both Lukes' and Hayward's approaches to power in a singular model encounter difficulties because they each have different ontologies that inform their conceptions of power [35]. Lukes and Hayward [6] both agree that conceptualizing power must go beyond overt exercises of power to also include the less observable—that cultural power as a form of power exists. Their disagreement is about who or what holds and exercises cultural power—about its source(s).

If the two are simply both included together as part of a "family resemblance" model [8,28], the result bypasses the actual theoretical conflict between Lukes and Hayward, often ending up with power definitions that are unclear or indiscrete regarding the forms of power they are describing and/or whether they are referring to actors, structures, or both. For example, Baker et al.'s [76] (p. 739) "instrumental power", meant to be actor-centered, is defined as "direct influence of one actor over another to affect decision-making and outcomes", but structures like laws and regulations exercise similar power (e.g., for schools, consider rules around public education access, educational standards, and building codes). Likewise, Baker et al.'s [76] (p. 739) "structural power", which includes such laws and regulations, is "a more diffuse form of power", more similar overall to 3D and de-faced power. While it includes such manifestations as "the shaping of informal norms", actors such as classroom teachers likewise shape informal norms, exercising and producing the content Baker et al. [76] (p. 739) designate as "structural".

An additive model reduces the utility of de-facing power by making it overly broad. In this case study, de-facing power can refer to state and federal special education laws that are coercive—public schools must comply or encounter legal consequences. This is closer in form to 1D than 3D power. Social norms around what is expected or celebrated versus boundary-pushing inclusion practices have a cognitive influence, closer to 3D power. To navigate these issues, definitions of power can combine forms of power together while separating out sources of power as a separate dimension that overlays these forms.

Power has multiple different dimensions that need to be considered. The form of power, e.g., 1D, 2D, and 3D power, is one dimension. Gaventa's [22] power cube also includes spaces and levels as other dimensions of power, where power, in any form, can also be at any space or level. For example, 3D power can operate at the global level or the household level, in closed spaces or claimed ones.

Not captured by the power cube model, the structure–agency continuum is another dimension warranting consideration. While Gaventa's [22] power cube conceptualizes forms of power through Lukes' actor-based model, 1D, 2D, and 3D power can each manifest as agential and as structural. De-faced power has elsewhere been referred to as "cultural power", recognizing sociocultural arrangements and practices as "a form of power" [26] (pp. 25,30). Three-dimensional power and de-faced power are both cultural power—the same particular *form* of power, but they differ based on their origins. Structural power operates at the 1D, 2D, and 3D levels in terms of the type of power being exercised. The source of power is another dimension of power to consider; one that can interact with any form (or space or level). The power cube's forms of power, like other typologies, are appropriate and avoid being problematic so long as they each describe discrete forms of power. The issue is not with their forms but with the lack of consideration of and differentiation by their origins.

Broad attempts to primarily define power by its source, giving source primacy over form, end up with a less powerful conceptualization and analysis of power's forms.

Hayward and Lukes [6] are united in their conception of cultural power as existing and important. However, their limitation is only considering cultural power from one vantage point—from one particular source. Instead, cultural power—indeed, all forms of power—can be evaluated and considered in relation to a variety of sources. Rather than choosing only the forest or the trees [71], power can be considered across its various multiplicative interactive agents. Lukes may prefer to conceptualize the structural dimension as "joint action" [65]. Hayward may believe our society's ideology of individualism warrants prioritizing a focus on how cultural power in the form of institutional boundaries impacts actors. Nonetheless, power must be considered across its various actors of power, from a more traditional actor/agent perspective, and from one in which structures, the "interlinkage of action" [65] (p. 59) also hold and exercise power.

Sources of power cannot be easily dichotomized into actor and structure. For example, the multiple interactive agents of power include the following: societal (amorphous and structural systems like norms, ideologies, meritocracy, individualism, racism, etc.); institutional (e.g., capitalism, education, institutions and structures); collectives (e.g., groups, particular institutional manifestations such as a particular school or organization or business); interpersonal (A to B and B to A); and individual (internalized). This initial sketch is offered not as a final typology of this dimension of power but as a flexible sample of the continuum of power's sources. These areas are overlapping and interrelated as well as ambiguous. Particular analyses may find it helpful to operationalize this dimension in a different way, such as breaking down collective categories into smaller categories so that a family and a school are at different levels of analysis.

Proceeding in this direction matches empirical reality, both as found in this case study and as already exists in stigmatization literature. Power has been accepted as being part of stigmatization's definition since Link and Phelan's 2001 article "Conceptualizing Stigma" [48], though what power means in relation to stigma is quite ill-defined in the literature. Stigmatization is well accepted as occurring at various different levels from various different sources, with models conceptualizing stigmatization that bypass the dichotomous agent–structural impasse. For example, Kumar et al. [81] (p. 630) conceptualize abortion stigmatization as having various nested "levels:.... Individual, community, organizational, structural, [and] framing discourses". Similarly, Pescosolido and Martin [82] (p. 92) note that there are various forms of stigmatization manifestation (e.g., "perceived, endorsed, anticipated, received, enacted"), but separately break down stigmatization sources: e.g., self-stigma is internalized by the self, public stigma is from "the general population", "provider-based stigma" is from "occupational groups designated to provide assistance to stigmatized groups", and "structural stigma,... also called institutional stigma", is located in "policies, laws, and constitutional practice". They also present the Framework Integrating Normative Influence on Stigma model, a "multilevel approach" that shows both "the individual" and "the community", the latter reflecting a "larger cultural context", as jointly contributing to the production of stigma [82] (pp. 87,102). Like with stigmatization literature, power can be conceived through various dimensions—including the dimension of sources.

## 6. Conclusions

This paper offers an empirical study that explicitly uses the constructs of both 3D and de-faced power to investigate whether they both seem to exist, and if so, whether considering them in tandem and relation to one another matters. Educators promote inclusion through various means at Mitchell School, including through the deployment of cultural power. However, while with more normative actions they frequently explain rationales to students, that metacognitive talk evaporates with more transgressive acts. Within this case study, both structural and agential power are evident, and considering

both sources of power and their interplay resulted in a more meaningful understanding of social phenomena, revealing insights that otherwise might have remained uncovered.

Hayward and Lukes [6] exchange bouts about what would be lost by not considering their own analytic approach to power or by only considering the other's analytic approach. They "talk past each other", a common characteristic of structure–agency discourse [42] (p. 58). While their paper format made for an enjoyable read and helped elucidate their arguments, despite its title, they engaged more in debate than "dialogue" [6] (p. 5) and [83]. This paper explored one possible manifestation that could result from Lukes and Hayward instead talking to and with each other, "with the purpose of learning more truth about the subject" (how to best conceptualize power) through their joint communication [83]. While Hayward and Lukes [6] focus on a debate they implicitly set up as zero-sum, this case study demonstrates that both Hayward's structural power and Lukes' agential power have empirical validity, and, furthermore, that there is something lost from not considering both structural and agential sources of power as well as their interaction. Power has multiple dimensions, one of which is the spectrum of sources, from individual actors to social structures, that can and do bring about effects.

Considering power across agential and structural vantage points broadens and improves analysis. For example, if an item fails to be on the agenda at a school board meeting, this would entail considering the agenda-maker's exercise of 2D power, the institutional biases and contextual conditions that influenced that agenda-maker, and their interplay. This has direct applied implications. Hayward discusses a practical example of the difference in discrimination outcomes in the U.S. judiciary that result from a focus on actor-centered power over de-faced power, noting that while under the Warren Court the threshold for discrimination was "discriminatory *effects"*, under the Burger and Rehnquist Courts, the threshold required intentional discrimination by an agent(s) [6] (p. 18). While I agree that Hayward is correct that courts ignoring systemic racism is problematic, the answer does not seem to be to *also* ignore explicit racism by landlords. A joint model could allow for considering both interpersonal and institutional discrimination and therefore also address Hayward and Lukes' [6] concerns about responsibility and accountability. Hayward indeed does not argue against holding actors accountable, responding to this very issue by stating that "If you are enacting the unjust structure, you can certainly be held blameworthy" (Clarissa Rile Hayward, communication during Q&A, September 18, 2022, full paper panel "Theorizing Structural Injustice and Responses to It", *American Political Science Association Annual Meeting & Exhibition*). If actors can be worthy of blame, meaning responsible and accountable for their actions, then, as Lukes argues, "the most appropriate explanation will be in terms of power" [6] (p. 11). If actors were simply structural dupes (Lukes labels individuals without agency as "powerless") [6] (p. 12), then they could not be "blameworthy".

Hayward can be right that there needs to be a societal shift towards prioritizing a focus on the institutions and practices that produce contexts important to freedom and equity [6], but it does not follow from this that actors cannot also be considered as having or exercising power. At Mitchell, broader norms and pressures both led the school to have a lesson centering transgender identity, and to a parent challenging this lesson and national right-wing attention being foisted on the school. However, individual agency is very apparent in this case. The guidance counselors decided to do the lesson because of their commitment to equity and because there were multiple transgender students at the school. While encountering vulgar backlash from across the country, the principal focused on how to improve communication but dismissed any pressure to end such programming. The principal's perspective had been informed in part in response to the murder of 23-year-old Charlie Howard in Maine in 1984, an anti-gay hate crime against a young man who was effeminate and openly gay. In some other schools, transgender students are met with hostility from educators rather than inclusion programming, and school administrators change course substantially in reaction to national attention directed at their school.

Lukes in turn may not discern a reason to label structures as powerful if they cannot be held accountable. But if power and accountability are linked, naming structures' roles can contribute to requiring actors' remediations address structural power. If a state guarantees equal educational opportunity, but that education is classist, racist, sexist, and ableist, considering the power of these processes means the state may be required to proactively tackle these processes. If hiring processes or bank lending processes are racist, then remedies may not simply be to end active discrimination but to set quotas companies are accountable for that require them to proactively challenge oppressive systems they were upholding. Without discussion of disembodied norms, powerful actors may be held accountable, but without being required to tackle any of the root causes that are producing inequality, creating a cycle of holding actors responsible without addressing the wellspring that produces these actors or the actual determinants of the inequality they are producing. In this case study, teachers may be powerful, but may not be aware of how they situationally exercise their power as a result of disembodied norms. They may be aware but feel powerless, with school administrators holding power to ensure teachers feel secure and supported in discussing inclusion robustly, and have the cultural competence to do so. School administrators may also feel powerless to navigate this within the current political environment and depending on their broader communities' political views, but here as well, identifying the underlying causes means administrators can elucidate the areas that produce discomfort and connect with organizations that can collaborate with them on how to navigate discussing and framing issues that may seem precarious or potentially problem-producing.

In this case study, an analysis that includes both actor-centered and de-faced sources of power reveals what in some cases may not be a fully conscious process or may represent siloed teacher practices. Awareness could generate increased agency, enabling educators to more intentionally navigate their school environment by recognizing what is happening and what choices they have, share their choices and rationales with their colleagues, and identify professional development needs. A power analysis without structural power limits their understanding, while a power analysis without giving them agency disables them. An analysis that considers the multiple interactive agents of power, from individual actors to social structures, enables recognition of these processes and their implications, and enables educators to make intentional decisions about how they want to navigate and be part of the system, about what they need from their institutions, and about whether or not to include metacognitive talk, and in what forms, in various situations.

Power analyses that aim to challenge oppressive power [10] would benefit from including the dimension of sources of power in their tools. For example, Gaventa's [22] power cube, meant as a tool to help unveil power in order to challenge it, could be expanded to include not just levels of power (from household to global) but also sources of power (from internalized to societal). These are separate concepts—e.g., at the national level, there are individual actors, group actors, norms, laws, etc.

Given the importance of power to social research, what else might we learn about the social world and about power itself if research repositions Hayward and Lukes' [6] conceptualizations of power forward from debate into collaboration? This case study provides one empirical example of how considering the interplay of structure (de-faced power) and agency (3D power) can produce a more meaningful understanding of social phenomena. "Social scientists frequently... repeatedly prioritize one single dimension or source [of power]", while neither is more important than the other [28] (p. 4). What might a both/and approach reveal theoretically about power and in turn about our social world? Hayward and Lukes [6] should re-enter their dialogue and engage in a deliberative thought exercise where they start from assuming that what the other person is describing is also power, and then consider from there how to make sense of power, how the other's conceptualization might enrich and move their ideas on power forward. What might we learn by conceptualizing and studying power in a manner that understands agency and

structure as co-occurring and continuously producing one another, and that accounts for the multiple interactive agents of power, from individual actors to social structures?

**Funding:** This research received no external funding. The APC was funded by Southern Illinois University Edwardsville.

**Institutional Review Board Statement:** The study was conducted in accordance with the Declaration of Helsinki, and approved by Institutional Review Board for the Protection of Human Subjects, University of New Hampshire (protocol code IRB #6407 and date of 22 February 2016), and the Institutional Review Board, Southern Illinois University Edwardsville (protocol code IRB #1844 and date of 15 November 2022).

**Informed Consent Statement:** Participant consent (and parental/guardian consent with student assent) were waived for the observational study. Caregivers and students were informed of the visitor policy through the Student-Parent Handbook, permission to access the building was given by the principal, and all district and school protocols were followed. Waiver of consent reflected no more than minimal risk to participants, no adverse impact on the rights and welfare of participants, the inability of the research to be practicably carried out without the waiver, and not practicing deception. For the follow-up interviews, informed consent was obtained, though obtaining signatures was waived to reduce logistic burdens on participants and because there was minimal risk to participants. Interview participants received a research participant notification form prior to participating in the study.

**Data Availability Statement:** Data is contained within the article. Field notes will not be shared to ensure confidentiality of participants' identities.

**Acknowledgments:** Thank you to Michele Dillon for guidance with conducting ethnographic fieldwork, Jordan Burke, Henry Herndon, and Emily Whitmore for coding and analysis collaborations, Mark Haugaard for feedback on my theorizing of power, and the educators and students at Horace Mitchell Elementary who welcomed me into their school.

**Conflicts of Interest:** The author declares no conflict of interest.

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
