# Peer review of "Evaporating Metacognitive Talk: School Inclusion, Power, and the Interplay of Structure and Agency"

_education, doi:10.3390/educsci14030320_

Round 1

Reviewer 1 Report

Comments and Suggestions for Authors

1. This article presents an empirical study that explicitly uses the constructs of both 3D power and counter power to examine whether both appear to exist and, if so, whether to consider them in tandem and relational issues. Research findings showed that teachers promote inclusion in a variety of ways at Mitchell School, including through the use of cultural strength.

2. The topic is original. Both structural and agential power are evident in the case study presented, and consideration of both sources of power and their interplay has resulted in a more meaningful understanding of social phenomena, revealing insights that may otherwise have gone undiscovered.

3. The methodology to use in the research is enough.

4. The conclusions are answers for main questions.

5. The references and tables are appropriate.

Author Response

Thank you for reading, evaluating, and reviewing my manuscript.

Reviewer 2 Report

Comments and Suggestions for Authors

Evaporating metacognitive talk: School inclusión, power, and the interplay of structure and agency.

The topic it brings is a topic of interest for the improvement of a more inclusive school, but it presents a number of methodological and substantive problems that prevent it from being published.

INTRODUCTION

The article states at several points that it is an empirical examination. What is your definition of empirical?Are empirical observation instruments used? Is an empirical test conducted? Where are these data?

The research questions are very ambitious: what is the culture of the Mitchell school, what is its climate, how does it foster inclusion, how does it reproduce inequalities, what theories of power are manifested in the school's inclusion dynamics (Lukes' theory or Hayward's theory). The questions are general, poorly  specified.

Do these questions in the Material and Methods section relate to the questions in section 2.3? There is a disconnect between these two moments of setting research questions. The meaning of section 2.3 is not understood.

It is stated that the central theme is inclusion, but the introduction to the article does not specify the theme of inclusion, it only makes a small reference to bullying and stigmatisation, but does not talk about inclusion in general. This leads to a lack of understanding of the dimensions of inclusion that he is going to work on by applying Lukes and Hayward's theory of structure and agency.

There is a very broad introduction to the structure and agency theories of power, but the introduction says very little about power relations and inclusion. It is not clear what kind of inclusion he is referring to, whether it is inclusion of race, gender, social status, etc.

MATERIAL AND METHODS

There are a number of questions related to this section of material and methods

Why is this school selected? What are you looking for in the observation?

You say you made 13 visits. Why this number of visits? What were you looking for in these visits?

Why do you visit one class more than once and others are not visited?

Who is observed, where, for how long, in which situation, which person?.

Why do you interview, in what situation, for what?

When taking notes, what kind of records do you use. Are they/are they not behavioural records, are you looking for something specific? Is the observation system open or closed?

For what purpose is the informal interview conducted. Explain the purpose of the interview in this case.

In relation to the role of the observer. Why do you go unnoticed by some and for others act as a researcher doing interviews, what role did you adopt, although you say you did not have a defined role, how did you identify yourself as a researcher, what kind...?

Was participant observation carried out, did the teachers and students know that they were being observed?

How was the observation recorded?

What coding system did you use? The coding system used should be shown.

What material is coded? What coding procedure was used? Deductive - Inductive - You start from a previous coding system based on experience.  Indicators for collecting data on inclusion and power are not identified.

It refers to an intersection of codes. Where is this intersection? Intersections of codes should be shown.

RESULTS

The examples he gives to illustrate the results are few and to some extent not specific to the issue of inclusion. They are examples of norms in which power is expressed in its agential and structural features but their association with the issue of inclusion is not clearly perceived,

The examples given in the results are very limited. The results do not provide clarifying results on inclusion and power. The results do not give specific data on power and inclusion and how they interact with each other. There should be a table or matrix representative of the results obtained.

The reference to metacognition is not clear where it is located, whether in structure, agency, power, inclusion....

DISCUSSION

The comments in the discussion are in some cases not confirmed by the data presented in the results.

For example, the deductions you make about instrumental and discursive power, where do they come from? From what results/data is the discussion on this issue deduced? 

CONCLUSIONS

How can you say in the conclusions that the empirical validity of the two positions of power is demonstrated, where are these data?

The article provides information on specific aspects of power structure and agency carried out in the Mitchell school, but not on their relationships and interactions. The conclusions it draws are not justified by the data.

Finally, the title and keywords should be reworded.

Reviewer 3 Report

Comments and Suggestions for Authors

My comments for authors are included in the word file attached below.

Round 2

Reviewer 2 Report

Comments and Suggestions for Authors

The new edition of the article responds, to a considerable extent, to the questions raised in the review. 

My recommendation is to publish the article

Reviewer 3 Report

Comments and Suggestions for Authors

Thank you for thoroughly attending to my earlier feedback on your manuscript, I am now confident in this paper being published without further amendments. Most particularly, the methods section now more explicitly identifies the decision-making processes around access to the research site and what was involved in the inter-rater reliability process. Your argument about the loss of meta-cognitive teacher talk and how this relates to power has also been foregrounded, carving out your contribution more clearly.